# IN DEFENSE OF WASSERSTEIN

## ABSTRACT

Since the introduction of Wasserstein GANs, there has been considerable debate whether they should be viewed as minimizing the Wasserstein distance between the distributions of the training images and the generated images. In particular, several recent works have shown that minimizing this Wasserstein distance leads to blurry images that are of much lower quality than those generated by commonly used WGANs.

In this paper we present theoretical and experimental results that suggest that with commonly used parameter settings, WGANs *do* minimize the Wasserstein distance but the form of the distance that is minimized depends highly on the discriminator architecture. We focus on discrete generators for which the Wasserstein distance between the generator distribution and the training distribution can be computed exactly and show that when the discriminator is fully connected, standard WGANs indeed minimize the Wasserstein distance between the generated images and the training images, while when the discriminator is convolutional they minimize the Wasserstein distance between *patches* in the generated images and the training images. Our experiments indicate that minimizing the patch Wasserstein metric yields sharp and realistic samples for the same datasets in which minimizing the image Wasserstein distance yields blurry and low quality samples. Our results also suggest alternative methods that directly optimize the patch Wasserstein distance without a discriminator and/or a generator.

## 1 INTRODUCTION

In a seminal paper, Arjovsky et al. (2017) showed the relationship between generative adversarial networks (GANs) and the Wasserstein distance between two distributions. They argued that when the data lies on a low dimensional manifold, the Wasserstein distance is a more sensible optimization criterion compared to the KL divergence and showed that the Wasserstein distance can be approximately optimized using an adversarial game between two neural networks: a generator network and a critic network. The key difference between their method, Wasserstein GAN (WGAN), and previous GANs is that the critic is regularized to be 1-Lipshitz, and a great deal of subsequent research has focused on improved regularization techniques Gulrajani et al. (2017); Miyato et al. (2018); Anil et al. (2019). WGANs have been used in many applications and can give excellent sample quality on different challenging image datasets Radford et al. (2015); Isola et al. (2017); Brock et al. (2018); Karras et al. (2020); Sauer et al. (2022); Pan et al. (2023).

In recent years, however, the connection between the success of GANs and the Wasserstein distance has been questioned Stanczuk et al. (2021); Fedus et al. (2018); Mallasto et al. (2019); Korotin et al. (2022). A first criticism is the extent to which WGANs indeed minimize the Wasserstein distance (W1). Several authors have shown that the approximating W1 using WGANs can yield a poor approximation and that when the approximation quality is improved, the quality of the generated images actually decreases Mallasto et al. (2019). A second criticism is whether minimizing the Wasserstein distance between two distributions is a sensible optimization criterion for generative models of images. Figure 1 is a reproduction of a figure from Stanczuk et al. (2021) showing that a mini batch of blurred images (Kmeans of the data) is *closer* to the entire dataset than a batch of real images in terms of Wasserstein distance. This suggests that even if we improved how well GANs approximate the Wasserstein distance, we would only decrease the quality of the generated images. This had lead to an alternative view whereby "GANs succeed because they fail to approximate the Wasserstein distance" Stanczuk et al. (2021) and that GANs should not be seen as minimizing a loss

function Goodfellow et al. (2020); Fedus et al. (2017). Many recent papers had completely deserted the distribution matching approach focusing on analyzing the adversarial game and its equilibrium Sidheekh et al. (2021); Farnia & Ozdaglar (2020); Schäfer et al. (2019); Qin et al. (2020).

While results such as those shown in figure 1 are certainly compelling (see Appendix A for more details), we believe that they are not sufficient to conclude that "GANs succeed because they fail to approximate the Wasserstein distance". First, these results are based on computing the Wasserstein distance between *batches* and not the Wasserstein distance between distributions. Computing the distance between small batches may not give a good estimate of the distance between the distributions. Second, given that GAN training is not solving a convex optimizatoin problem, it is hard to draw conclusions from the solution obtained in a particular run of training GANs. One can imagine that a particular run converged to a local minimum of the Wasserstein distance but another run will give a better loss.

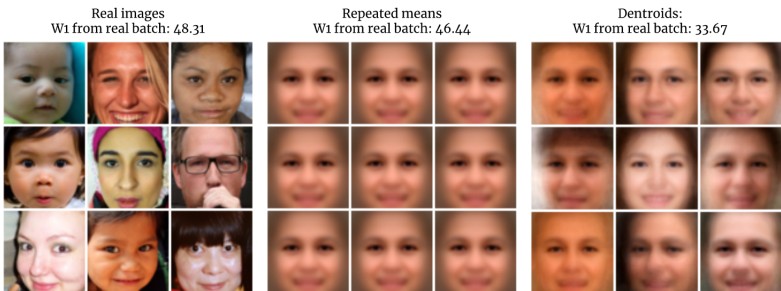

Figure 1: A recreation of Figure 10 in Stanczuk et al. (2021). 3 minibatches of 64 images are compared to the same batch of 10000 real images. Left: a minibatch of real images. Middle a batch of repeated means of the data. Right: 64-means of the data. The two blurred batches give *better* Wasserstein distance than samples from the training set.

In this paper we present theoretical and experimental results that suggest that with the appropriate parameter settings, WGANs *do* minimize the Wasserstein distance but the form of the distance that is minimized depends highly on the discriminator architecture. We focus on discrete generators for which the Wasserstein distance between the generator distribution and the training distribution can be computed exactly and show that when the discriminator is fully connected, standard WGANs indeed minimize the Wasserstein distance between the generated images and the training images, while when the discriminator is convolutional they minimize the Wasserstein distance between *patches* in the generated images and the training images. Our experiments indicate that minimizing the patch Wasserstein metric yields sharp and realistic samples for the same datasets in which minimizing the image Wasserstein distance yields blurry and low quality samples. Our results also suggest alternative methods that directly optimize the patch Wasserstein distance without a discriminator and/or a generator.

## 2 THE DISCRETE WASSERSTEIN PROBLEM

We start by reviewing the connection between Wasserstein distance and WGANs. The Wasserstein distance $W(P, Q)$ between two distributions is defined as:

$$W(P, Q) = \inf_{\gamma \in \Pi(P,Q)} E_{x,y \sim \gamma} \|x - y\| \tag{1}$$

where $\Pi(P, Q)$ denotes the set of joint distributions whose marginal probabilities are $P, Q$. The connection to GANs is more evident in the dual form:

$$W(P, Q) = \max_{f \in F_1} E_P(f) - E_Q(f) \tag{2}$$

where $F_1$ is the class of 1-Lipschitz bounded functions. Thus if we denote by $P$ the distribution over images defined by the generator and $Q$ the training distribution, the minimization of $W(P, Q)$ can be performed using an adversarial game in which the the generator attempts to decrease $E_P(f) - E_Q(f)$ and the discriminator, or critic, attempts to increase $E_P(f) - E_Q(f)$ where $f$ is the discriminator.

When $P, Q$ are continuous distributions in high dimensions, neither the primal nor the dual form can be computed exactly. This difficulty of exact computation makes it difficult to say whether WGANs indeed minimize the Wasserstein distance. One option is to look at small batches of images from $P$ and $Q$ and compute the average distance between such batches (see figure 1 for an example). However, as we show in appendix A (see also Bellemare et al. (2017)), the results may be quite different depending on the size of the batch.

In order to enable exact computation of $W(P, Q)$ we focus on the case where both $P$ and $Q$ are discrete distributions. In GAN training, the training set $Q$ is always discrete since it consists of $N$ training examples. We can also force $P$ to be discrete by first sampling $M$ fixed latent vectors $z_i$ from a standard normal distribution and then defining the following generative model.

- Sample $i$ from a uniform distribution over $1, \cdots M$.

- $x = f_\theta(z_i)$.

Where $f_\theta(z)$ is a standard generator as is used in all GANs. The only difference between this generative model, which we call "discrete GAN" is that the latent vector $z$ can take on at most $M$ possible values. Thus we can write the generated distribution $P$ as $P = \{x_j\}_{j=1}^M$ with $x_j = f_\theta(z_j)$ and the training distribution is $Q = \{y_i\}_{i=1}^N$. Then:

$$W(P, Q) = \sum_{i,j} \pi_{ij} \|y_i - x_j\| \tag{3}$$

with $\pi_{ij}$ a $M \times N$ matrix that can be computed using optimal transport in time that is polynomial in $M$ and $N$. In our expereimtnts we used the POT package Flamary et al. (2021).

Not only does the discrete setting allow us to compute $W(P, Q)$ exactly it also yields a simple algorithm for minimizing $W(P, Q)$, which we call "OT-Means".

**Algorithm (OT-Means).**

- Given $\{x_j\}$ set $\pi$ to be the optimal transport matrix between $\{x_j\}_{j=1}^M$ and the training set $\{y_i\}_{i=1}^N$

- Given $\pi$ minimize:

$$x_j = \arg\min \sum_i \pi_{ij} \|y_i - x_j\| \tag{4}$$

This minimization is the geometric median problem and can be performed using iteratively reweighted least squares. Weiszfeld (1937)

It is easy to show that this algorithm decreases $W(P, Q)$ at each iteration and it also allows us to characterize the optimal solution.

**Theorem 1:** Given a training set $P = \{y_i\}_{i=1}^N$, the set of $M$ vectors $Q = \{x_j\}_{j=1}^M$ that minimize $W(P, Q)$ is obtained when each generated sample is a linear combination of at least $N/M$ training samples. When $N = M$ then the optimal solution is simply $x_j = y_i$.

**Proof sketch:** This follows from the fact that the OT means is guaranteed to decrease $W(P, Q)$ so that the global optimum must be a fixed-point of the algorithm and hence also be a fixed point of the iteratively reweighted least squares algorithm which means that it is a linear combination of the training images. (see B for more details)

To illustrate Theorem 1, we show in figure 2 (second column) the (locally) optimal solution to $W(P, Q)$ computed using OT-means for three datasets with $M = 64$ The datasets are a toy dataset of squares at different locations, MNIST and FFHQ Karras et al. (2019). In all three cases $N >> M$ so that theorem 1 implies that the optimal solution is obtained when each generated sample is a combination of many training examples, and as expected the optimal solution generates blurred, unrealistic samples. These results agree with previously published Stanczuk et al. (2021) but here we can characterize the minimum and show that the optimal solution for the full Wasserstein distance should produce blurred images.

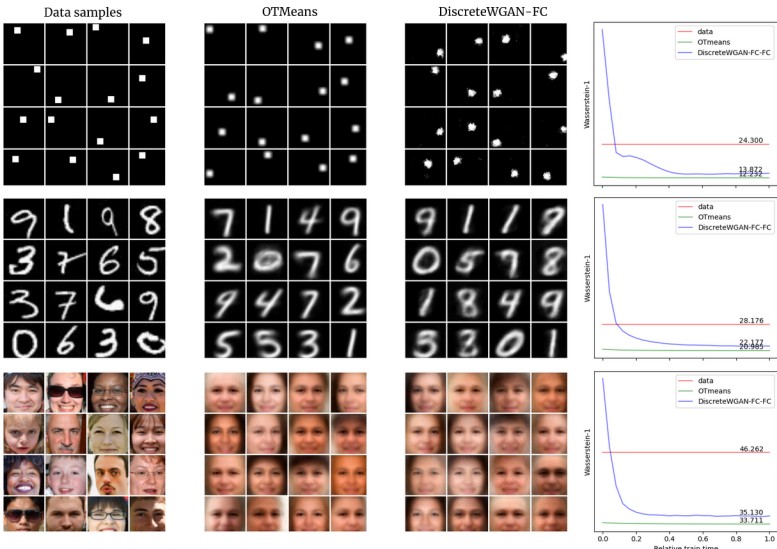

Figure 2: A comparison of OTmeans and a discreteWGAN with FC discriminator. Both generate exactly 64 images. The first three columns show 16 samples from the train data and the two learned models. The right column shows the Image-level W1 distance throughout the learning process. The distance between a batch of real samples and the dataset is added for reference in red.

## 2.1 WGANs WITH FULLY CONNECTED DISCRIMINATOR MINIMIZE THE WASSERSTEIN DISTANCE

The third column of figure 2 shows the result of running a discrete WGAN on the three datasets. The generator is a fully connected neural network with 1 layer with 128 hidden units, and the latent vector $z$ is chosen from one of $M = 64$ possible random vectors. The discriminator is also a fully connected neural network with 1 layer with 1024 hidden units. As is common in WGAN training we do not train the discriminator until convergence but rather do a single update after each mini-batch and use the gradient penalty to regularize the generator.

As can be seen in the figure, in all three datasets, the WGAN consistently decreases the Wasserstein distance during training and converges to a value that is close to the (locally) optimal solution found by OT-means. Not only are the losses obtained by WGAN and OT-means similar, the results are visually similar: both the optimal solution and the WGAN solutions are blurred.

Figure 9 in appendix C shows the result of the discrete WGAN when $M = N$. In this case, the global optimum of the Wasserstein is to copy the training examples which would give a loss of zero. As can be seen in the figure, the WGAN indeed copies the examples (even though the generator never sees the training examples directly) and obtains a loss that is close to zero.

To summarize our results so far, we find that when the discriminator is fully connected neural network, WGANs *do* minimize the Wasserstein distance for highly different datasets even when we use the standard training procedure in which the discriminator is not trained until convergence. We have also shown that for a discrete generator with $M << N$ the optimal solution in terms of Wasserstein distance will generate blurred images. How then do we explain the fact that WGANS used in practice do not generate blurred images?

## 3 WGANs WITH A CNN-GAP DISCRIMINATOR MINIMIZE THE PATCH WASSERSTEIN DISTANCE

WGANs used in practice do not use a fully connected discriminator. Rather, the overwhelming majority of them use a CNN architecture for the discriminator, similar to the architecture that is used for visual recognition. We now ask, how does this choice of architecture effect the relationship between WGANs and the Wasserstein distance?

We first focus on a CNN architecture in which the penultimate layer performs global average pooling. In such architectures, which we call CNN-GAP, a series of convolutions over $C$ channels is performed to produce $C$ spatial maps. Each spatial map is then averaged to produce a vector of length $C$ and the output of the network is a linear function of this vector. An example of a CNN-GAP architecture is the popular ResNet He et al. (2016) architecture that for many years was the standard CNN used for visual recognition and has also been used in many successful GANs Miyato et al. (2018); Brock et al. (2018); Sauer et al. (2021). For such an architecture, we can show that WGANs approximate the Wasserstein distance between patches, not images.

**Theorem 2:** Consider training a WGAN where the critic is a CNN which performs global average pooling in the penultimate layer and the size of the receptive field of units in the layer before the global average pool is $S$. Then WGAN training is approximately minimizing the Wasserstein distance between $S \times S$ patches in the generated images and the training images.

**Proof:** We can write the output of the critic for an input x as:

$$f(x) = \sum_{c=1}^{C} w_c \frac{1}{J} \sum_i f_c(P_i x) \tag{5}$$

where $P_i x$ extracts the $i$th patch in image $x$, $C$ is the number of channels, $w_c$ are the weights in the final layer, and $J$ is the number of patches.

We can define a patch critic, $g(x)$ as the function $g(x) = \sum_c w_c f_c(x)$ where $x$ is a patch. Then we can rewrite the image critic, $f(x)$ as:

$$f(x) = \frac{1}{J} \sum_i g(P_i x) \tag{6}$$

By the linearity of the expectation $E_P(f)$ is equal to $E_{\tilde{P}}(g)$ and likewise $\mathbb{E}_Q(f) = E_{\tilde{Q}}(g)$ where $\tilde{P}, \tilde{Q}$ is the distribution over patches in the generated images and the training set. This means that:

$$\max_{f \in GAP} E_P(f) - E_Q(f) = \max_{g \in G} E_{\tilde{P}}(g) - E_{\tilde{Q}}(g) \tag{7}$$

where $GAP$ is the class of functions that can be implemented by a CNN-GAP architecture and $G$ is the class of functions that operate on $S \times S$ patches and can be implemented by the units in the layer before the GAP.

The term "approximately minimizing" in the theorem statement is the same as in the original proof of WGANs. For the WGAN training approximation to be exact, the discriminator needs to be trained until convergence and the gradient penalty regularization should be equivalent to constraining the generator to be 1-Lipshitz. In practice, neither of these conditions hold. Nevertheless, we find that when the standard WGAN training protocol is employed, WGANs with a CNN-GAP discriminator indeed minimize the patch Wasserstein distance. The Middle images column of figure 3 shows the result of training a discrete WGAN with a CNN-GAP discriminator on the three datasets. We used 3 convolution layers with 64 channels, kernel size of 3 and stride 2 followed by a GAP and a linear unit. Therefore its receptive field size is 15. We used exactly the same generator as in figure 2 (replotted here in the leftmost column) and exactly the same training protocol. The only difference is the architecture of the discriminator and this yields a dramatic change in the generated images. For the Fully-Connected architecture the generated images preserve the global structure but not the local statistics, while for the CNN-GAP discriminator the generated images are sharp and preserve local structure but not the global one.

Even when the generator is discrete, the Wasserstein distance between all $16 \times 16$ patches in the generated images and the training set is too expensive to compute. As a cheaper proxy, we compute the sliced Wasserstein Distance (SWD)Pitie et al. (2005); Rabin et al. (2011); Bonneel et al. (2015); Kolouri et al. (2018); Deshpande et al. (2018); Elnekave & Weiss (2022) between the sets of patches. Given a candidate set of images $\{x_j\}$, we extract all patches from this set and all patches from the training images $\{y_i\}$ and measure the SWD between the two sets of patches. We can also approximate the optimal solution to the Patch-Wasserstein problem by optimizing the SWD with respect to the images $\{x_j\}$ using gradient descent. As shown in the third column of figure 3, the results of direct optimization are visually similar to what is found by a WGAN with a CNN-GAP

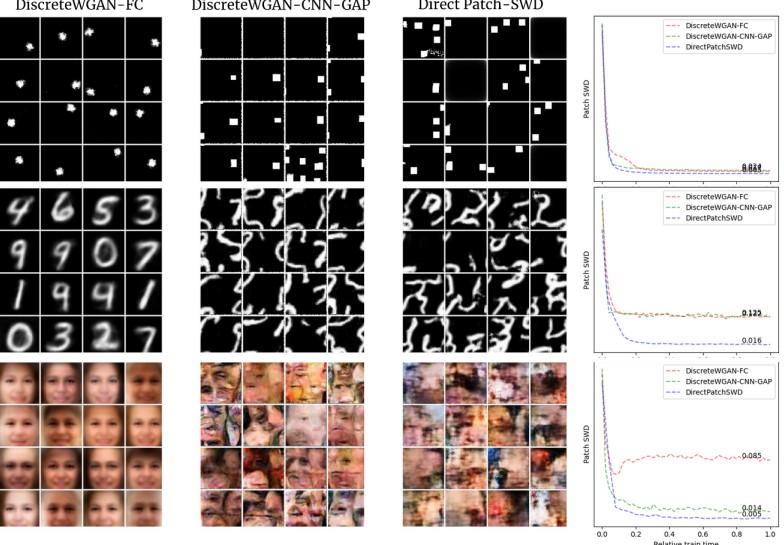

Figure 3: Training the same FC generator with different discriminator architecture gives a dramatic change in the generated images. Fully connected discriminator (left column, replotted from figure 2 yields blurred images. CNN-GAP discriminator (second column) gives sharp images but without global structure similar to directly minimizing the patch Wasserstein distance (third column). The SWD loss over patches (right column) is decreased during training with a convolutional architecture and approaches the loss of direct minimization.

discriminator: the generated samples preserve the local statistics but not the global ones. As shown in figure 3 the patch SWD distance decreases steadily during training of the WGANs with a CNN-GAP discriminator and approaches the result of directly optimizing it.

## 4 PATCH WASSERSTEIN VS. IMAGE WASSERSTEIN

Theorem 2 shows that when WGAN is trained with a convolutional discriminator, the WGAN generator is approximately minimizing the Patch Wasserstein metric rather than the image Wasserstein metric. But is there any theoretical reason to prefer minimizing the Patch Wasserstein rather than the image Wasserstein? The following theorem shows that the situation of figure 1, where the image Wasserstein distance prefers blurred samples over a batch from the training set, does not occur when we use patch Wasserstein with small patches.

**Theorem 3:** Suppose that the distribution over patches has finite support (i.e. there are at most $K$ possible patches in the training distribution). Suppose that $M < N$ and that the number of patches in the generated $M$ images is much larger than $K$. Then a sample of $M$ images from the test set will be closer in terms of *patch Wasserstein* distance to the training set of $N$ images than almost any $M$ images that are a linear combination of $N/M$ training examples.

**Proof:** Since the number of patches is large in the $M$ generated images and in the training set, and they come from the same distribution, then the Wasserstein distance over patches will be close to zero. On the other hand, when we take linear combination of $M/N$ different images we will almost always create additional patches that are not one of the $K$ possible patches in the distribution.

A trivial example where the assumptions of theorem 3 hold is the square dataset. Since each pixel in the distribution can only take on the values of 0 or 1, a $3 \times 3$ patch can take on at most 512 possible values (the actual number is actually much smaller). When we compare a sample of 64 images from the test set to the training set, then both patch distributions have at most 512 possible values and will have similar distributions over these 512 possibilities. In contrast, when we take the result of OT-means for this dataset, the patch distribution includes additional patches (with non binary pixel values) that do not occur in the training distribution. Thus patch Wasserstein will prefer a sample from the test set over the results of OT-means.

To test whether these assumptions also hold for more complicated distributions such as MNIST and FFHQ we repeated the experiments of figure 1 for the three datasets: squares, MNIST and FFHQ and compared the Wasserstein distance between 64 images from the test set and the full training set to the Wasserstein distance between 64 centroids of OT-means and the full training set. The result of these experiment is shown in the bar plot in Figure 4. In all three cases, the patch Wasserstein distance preferred the 64 images from the training set while the image W1 preferred the blurred centroids.

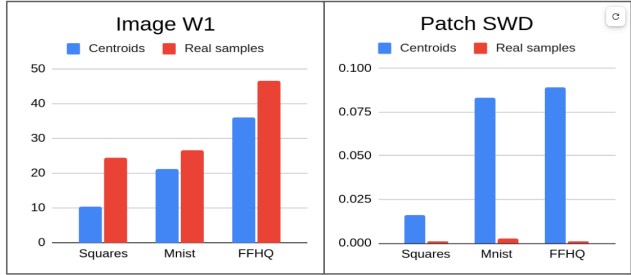

Figure 4: Comparing image and patch level Wasserstein distances. According to the image Wasserstein distance, a batch of blurred centroids is closer to the training disribution than a batch of the same size from the test set. With patch Wasserstein this is no longer the case.

However, as shown in the right image column in figure 3, simply optimizing the patch Wasserstein distance can yield images that have the correct patches but in the wrong location (e.g. eyes in the bottom of the image). This is not what we see with commonly used GANs that use a CNN discriminator but do not use GAP.

## 5 WGANs with a CNN Discriminator minimize the location-specific patch Wasserstein Distance

We now focus on a CNN architecture without global average pooling. We show that it minimizes a location-specific Wasserstein distance over patches.

**Theorem 4:** Consider training a WGAN where the critic is a CNN without a GAP and the size of the receptive field of units in the layer last linear layer is $S$. Then WGAN training is optimizing an upper bound on all location-specific Wasserstein distances between $S \times S$ patches in the generated images and the training images.

**Proof:** We can write the output of the critic for an input x as:

$$f(x) = \sum_c \sum_i w_{ic} f_c(P_i x) \tag{8}$$

where $P_i x$ extracts the $i$th patch in image $x$, $C$ is the number of channels and $w_{ic}$ the weights in the final layer.

Now define the function class $CNN$ as all functions that can be implemented using equation 8 and the class $CNN_i$ as the set of functions that can be implemented by equation 8 when $w_{jc} = 0$ for all $j \neq i$.

$$\max_{f \in CNN} E_P(f) - E_Q(f) \geq \max_{f \in CNN_i} E_P(f) - E_Q(f) \tag{9}$$

and by the same linearity of expectation as before:

$$\max_{f \in CNN_i} E_P(f) - E_Q(f) = \max_{g_i \in G} E_{\tilde{P}}(g_i) - E_{\tilde{Q}}(g_i) \tag{10}$$

where again $g_i(x)$ is a critic for the $i$th patch: $g_i(x) = \sum_c w_{ic} f_c(P_i x)$. Equation 10 is the Wasserstein distance between the distribution of the $i$th patch in the true images and the distribution of the $i$th patch in the fake images. Thus training a WGAN with a CNN discriminator minimizes an upper bound over all location-specific patch Wasserstein distances.

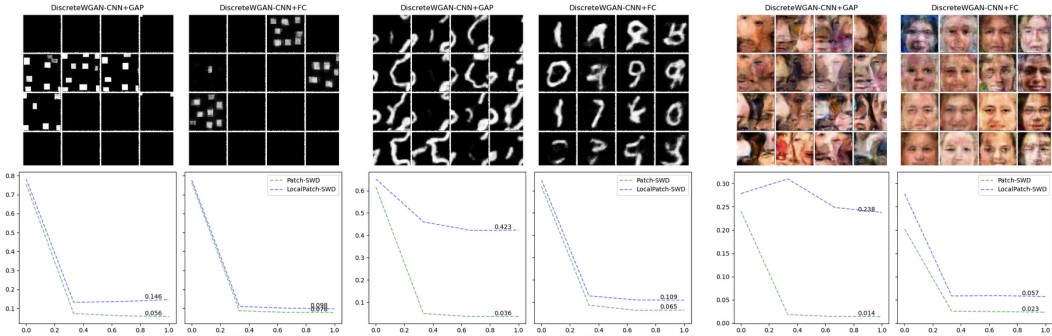

Figure 5: Changing the architecture of the final layer in the CNN discriminator causes a dramatic change in the generated images. When that layer has global average pooling followed by a linear function (CNN-GAP) local statistics are preserved but the location of features is not (e.g. eyes in the bottom of a face). When the final layer is fully-connected the location is preserved. The loss plots in the bottom show both CNN discriminators minimize the patch distribution distance to the train patches only the CNN-FC minimizes the location specific patch SWD to the training patches.

Figure 5 shows the results of training a discrete WGAN with two different CNN discriminator on the three datasets. The only difference between these results is the final layer in the discriminator: GAP followed by a linear layer (CNN-GAP) vs a single linear layer (CNN-FC). As predicted by theorems 2 and 4, the images are very different. In both CNN-GAP and CNN-FC the images are sharp but only in the CNN-FC the patch distributions that are being optimized are location-specific so that patches in the generated images appear in the correct location (e.g. there are no eyes in the bottom).

While CNN-FC discriminator cause the generated images to preserve the locaton of features, the images do *not* maintain the global structure. This is most evident in the squares dataset. While the location specific patch distribution is approximately preserved, the global structure is not (even though the training images always include exactly one square, some generated images have many squares and other none). Similarly, in the FFHQ results, while eyes always appear in the correct location, the symmetry of faces is *not* preserved and the two eyes in a generated face may be quite different (best seen by zooming in on figure 5). This is an inherent problem with using a fully convolutional architecture followed by a linear layer in the end. Such an architecture can only reweight the outputs of the convolutional layers and so cannot learn global structure that is much larger than the receptive field size. A fully-connected architecture, on the other hand, can learn global structure, but using it as a discriminator makes the WGAN minimize the image Wasserstein and hence produce blurred images.

The preceding discussion illustrates the challenge of choosing the correct discriminator for a given dataset. If the discriminator is fully-connected, global structure is preserved but local statistics are not. If the discriminator is convolutional with a receptive field that is too small, local statistics are preserved but global structure is not. As an alternative to a careful choice of discriminator architecture, one can simply choose to optimize *multiple* Wasserstein distances at different scales. Figure 6 shows one such algorithm. We start by running OT-means to optimize image level Wasserstein and then switch to optimizing SWD on patches. The results preserve both global structure and local statistics. See more details in appendix D.

## 6 LIMITATIONS AND EXTENSIONS

The main limitation of our analysis and our experiments is that there is a large gap between the GANs that we analyze and state-of-the-art GANs. There are many "tricks of the trade" that are important to get state-of-the-art image samples, including the choice of generator, working with pre-trained representations and different regularization techniques Sauer et al. (2022). Our focus here was on relatively simple GANs that still capture the essence of WGAN training and can show the strong effect of the discriminator architecture on the metric that is being optimized.

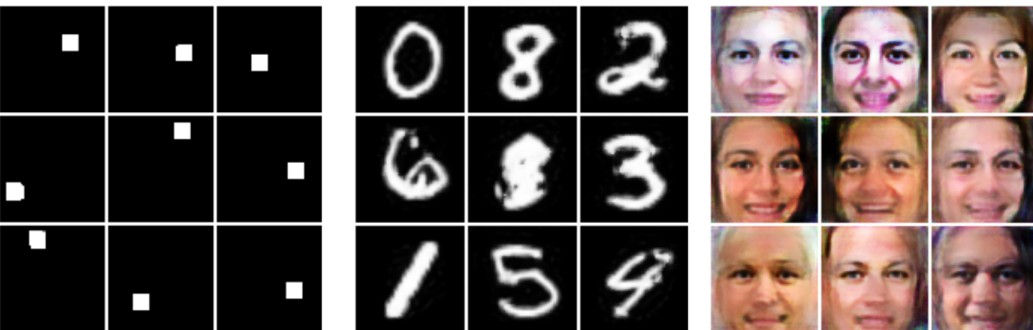

Figure 6: Results of direct patch SWD minimization in mulitple scales on the three datasets used throughout this paper. This generates novel samples that preserve both global structure and local statistics.

Another simplification that we made in the theory and the experiments is that of the discrete GAN. Clearly, without some form of capacity control on the generator, the minimal Wasserstein distance solution is to simply copy the training examples. We have found that with standard continuous WGANs that are trained with small training sets, overfitting does actually occur but as the size of the training set increases the WGANs no longer overfit. This suggests that even when the latent vector is continuous, WGANs have some implicit form of capacity control rather than the explicit form that we assumed in the paper.

Finally, our theoretical analysis in theorems 2 and 4 ignore all architectural constraints of the CNN except for the size of the receptive field in the final convolutional layer. Just as our theorems show that the specific form of the layers used in the last layers make a big difference in terms of the distance being minimized, there may also be an additional effect of the initial layers. In particular, the definition of the "receptive field" of a unit can be subtle. While methods such as ResNet have a theoretical receptive field that is almost the size of the full image, empirically it has been shown that the receptive field is much smaller (e.g. approximately 15% of the image size, according to Brendel & Bethge (2019)).

# 7 DISCUSSION

With the appropriate parameter settings, modern GANs can generate remarkably realistic images and yet the question of why they are so successful in some settings and not in other is still unsolved. The Wasserstein distance between distributions is a well-studied, principled way of comparing distributions and this is perhaps what makes the the link between WGANs and Wasserstein distances so intriguing. In recent years, two main arguments have been made against the connection between Wasserstein distance and successful GANs: (1) GANs do a poor job of approximating the distance and (2) minimizing this distance will lead to images of poor quality. In this paper we have attempted to address both arguments. We have shown theoretical and experimental evidence that WGANs do indeed optimize Wasserstein distances but the exact form of the distance being optimized depends on the architecture of the discriminator. When the discriminator is fully connected, WGANs minimize the image Wasserstein distance and produce images that are blurred. When the discriminator is a CNN with GAP, WGANs minimize the patch Wasserstein distance and produce images that are sharp but do not preserve the location of features. Finally, when the discriminator is a CNN with a final layer that is linear in channels and location, WGANs minimize an upper bound over location specific patch Wasserstein distances and produce images that are both sharp and preserve feature location.

Regardless of WGANs our results also indicate that minimizing the Wasserstein distance between patch distributions is a sensible optimization criterion that can lead to sharp, realistic samples. We hope that our paper motivates further research into finding good algorithms to optimize this distance.

## 8 REPRODUCIBILITY STATEMENT

We submit as part of the supplementary material a zip file containing the code we used for all the experiment in this paper. The code is written in python and a *README.md* file in the main directory explains how to use it as well as how to produce each and every of the paper's figures. The zip also contains instructions about preparing the data for these experiments. We highly encourage the interested reader to try out the code and see how different parameters affect the experiments results.

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

## A EFFECT OF BATCH SIZE

Figure 7 and figure 8 show how the results from 1 change with the size of the minibatch that is compared to the entire dataset. We repeat the experiments in these figures comparing sharp and blurry minibatches to the entire datasets, measuring the distance with image and patch level distances and mini-batch sizes growing from 10 to 1000. Figure 8 shows 9 samples from batches that are compared. The repeated mean stays and real images look the same for all minibatch sizes so we plotted 9 samples from them just once. The K centroids, which do change as K grows we show 9 samples from each minibatch. As can be seen in 8-a, for larger batch sizes repeated means are no longer better than sharp images while the centroid batches are consistently closer to the data then the real minibathces while still looking blurry. The picture change once we use patch-level losses and as can be seen in figure 8-b.

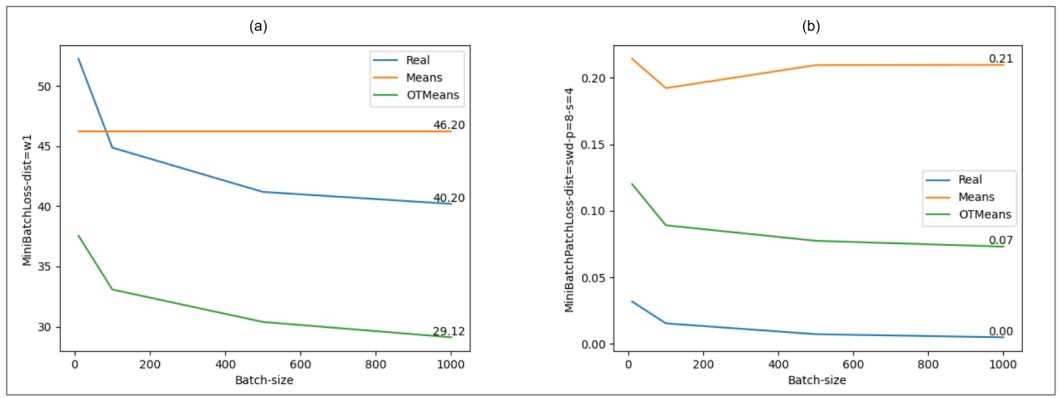

Figure 7: Effect of minibatch size on the Wasserstein distance between blurry and sharp batches to the real image dataset. (a) Image level wasserstein-1 distance. (b) Patch level SWD.

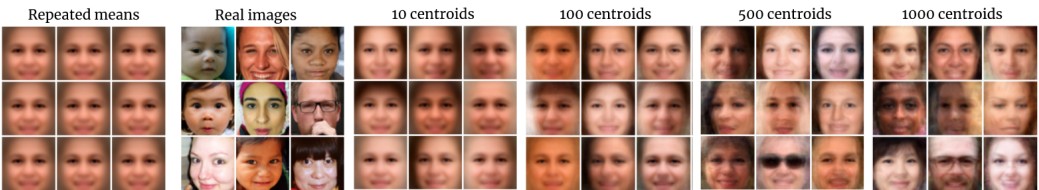

Figure 8: Samples from the minibatches used for the experiemnt in 7

## B OT-MMEANS

We present here a detailed description of the OT means algorithm as well more details about the proof sketch in section 2:

We consider here an arbitrary discrete generative model, only able to generate $M$ distinct images $\{x_i\}_{i=1}^M$. The goal is to minimize the W1 distance to $N$ train images $\{y_i\}_{i=1}^N$.

Algorithm 1 shows a detailed description of the OTmeans algorithm. In line 8 of the algorithm we makes use of the Weiszfeld algorithm Weiszfeld (1937); Beck & Sabach (2015) to solve $y_i \leftarrow$

$argmin_{z} \sum_{j=1}^{N} M_{i,j}||y_i - z||_2$. Note that as from the second iteration of the algorithm the estimated "centroids" are weighted sums of data points. The following theorem shows the distance in 11 non-increasing.

$$\underset{x_1..x_M \in R^d}{argmin} \ W1\left(\{x_i\}_{i=1}^{M}, \{y_i\}_{i=1}^{N}\right)() \tag{11}$$

**Theorem 1.** *If a solution for (11) exists then there is a solution of the form*

$$\forall_{i \in \{1,..,M\}} \ x_i = \sum_{j=1}^{N} w_j y_j$$

*where $w \in (\mathbb{R}^+)^N$*

*Proof.* Let $x_1^*...x_M^*$ be a solution to problem 11. Let $\Pi \in R^{NxK}$ an optimal transport map between $\{x_i^*\}_{i=1}^{M}$ and $\{y_i\}_{i=1}^{N}$. That is

$$W1\left(\{x_i^*\}_{i=1}^{M}, \{y_i\}_{i=1}^{N}\right) = \sum_{i=1}^{N}\sum_{j=1}^{M} \Pi_{i,j}||y_i - x_j^*||_2$$

Now define

$$\forall_j \ x_j' = \underset{z \in R^d}{argmin} \sum_{i=1}^{N} \Pi_{i,j}||y_i - z||_2$$

and since $\Pi$ is a valid (not necessarily optimal for $x_j'$s) transport map we can write:

$$W1\left(\{x_i'\}_{i=1}^{M}, \{y_i\}_{i=1}^{N}\right) \le \sum_{j=1}^{M}\sum_{i=1}^{N} \Pi_{i,j}||y_i - x_j'||_2 \le W1\left(\{x_i^*\}_{i=1}^{M}, \{y_i\}_{i=1}^{N}\right)$$

The right inequality is due to the definition of $x_j'$ as argmins and the left inequality is due to $\Pi$ being not necessarily optimal for $\{x_i'\}_{i=1}^{M}$ and $\{y_i\}_{i=1}^{N}$.

We now know that $x_1'...x_M'$ are optimal. From the definition of each $x_j'$ as the minimum of a weighted sum of distances to $N$ points we infer Weiszfeld (1937); Beck & Sabach (2015) that it is a non-negatively weighted sum of these points. □

---

**Algorithm 1** A pseudo-code for our OT-Means algorithm. At each iteration the optimal transport map between the current centroids $\{x_i\}$ and the train images $\{y_i\}$ is computed. Then the lines of the optimal transport map are used t

---

**Input**: Train images $\{y_i\}_{i=1}^{N}$, integer $M$ for number of images to optimize. $T$ number of Weizsfeld iterations.
**Output**: Optimized images $\{x_i\}_{i=1}^{M}$
0: **for** i=1,M **do**
0:     $x_i \sim N(0, \sigma I)$
0: **end for**
0: **while** not converged **do**
0:     $\Pi \leftarrow OT(\{x_i\}, \{y_i\})$
0:     **for** i=1,M **do**
0:        **for** j=1,T **do**    # Weiszfeld iterations (Weiszfeld, 1937)
0:           $x_j \leftarrow \frac{1}{\sum_{h=1}^{N} \frac{\Pi_{h,j}}{||y_h - x_j||_2}} \sum_{i=1}^{N} \frac{\Pi_{i,j}}{||y_i - x_j||_2} y_i$
0:        **end for**
0:

---

## C   OTMEANS AND DISCRETEWGAN WHEN M=N

The figure below is equivalent to 2 in the case where M=N=1000. OT means clearly copies the training examples and reaches zero loss. The discrete GAN seem to fail to reach zero loss but looking at the earest neighbor of the generated images as can be seen in 10 it clearly goes in the direction of copying the training samples only with some noise artefacts that are probably due to the noisy signal that comes from the discriminator. Note that the discrete GAN have only access to the discriminator and not to the training samples themselves.

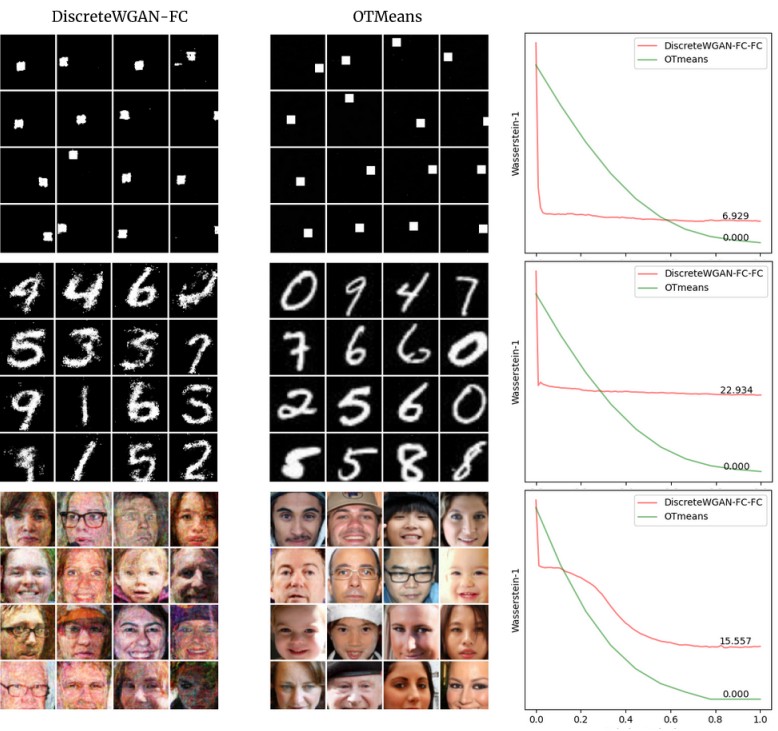

Figure 9: A comparison of OTmeans and a discreteGAN with FC discriminator. Both generate exactly 1000 images, the left and middle shows 16 samples of them. The right column shows the Image-level W1 distance throughout the learning process.

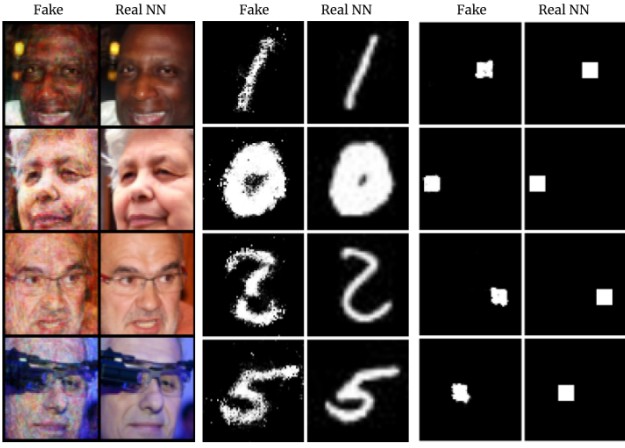

Figure 10: For 4 out of the 1000 learned images we show their nearest neighbor from the 1000 training samples. The learned images look like noisy copies

# D GENERATING SHARP IMAGES WITH MULTI SCALE DIRECT PATCH DISTRUBITON MATCHING

We bring here an additional figure to figure 6 that shows the nearest neighbor of some of the generated images from the train set. As can be seen the generated images are not exact copies of the data.

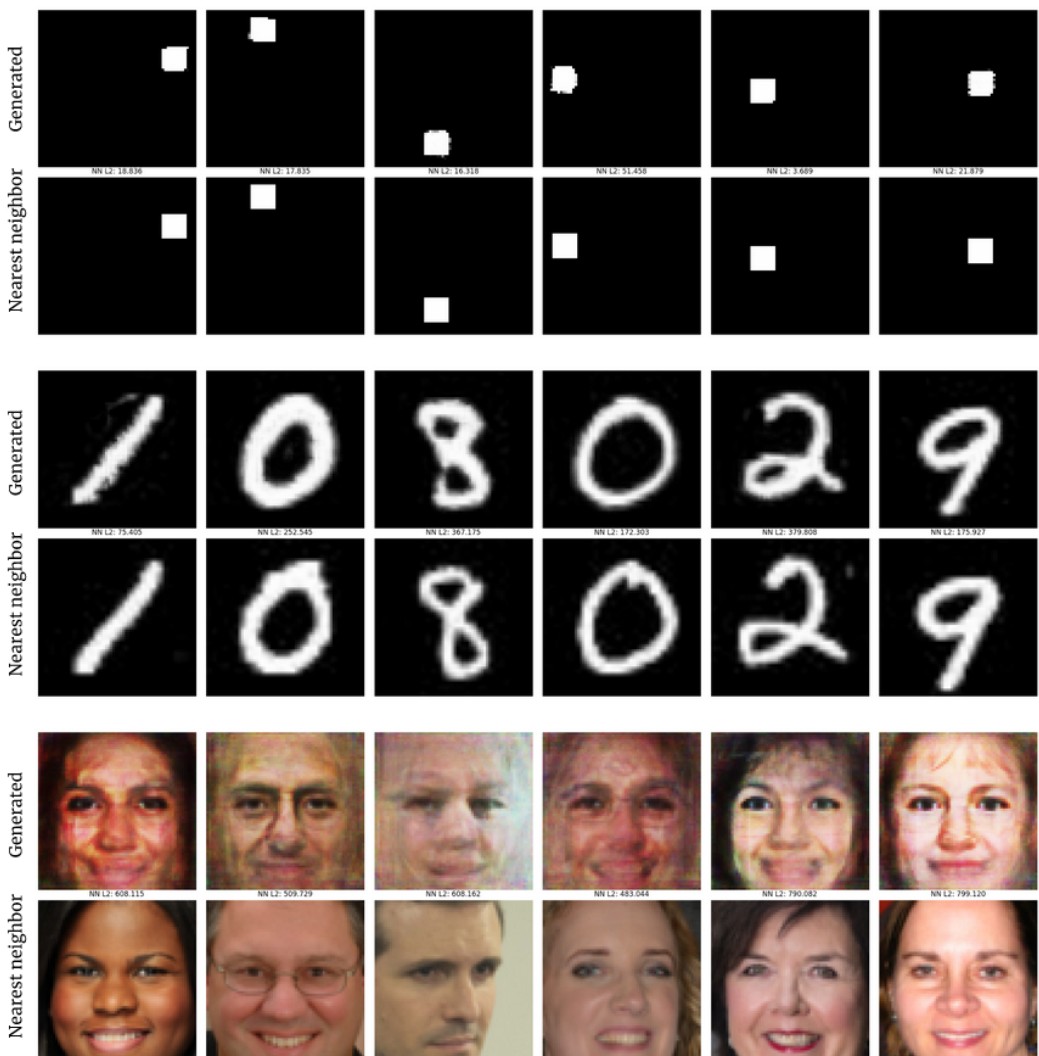

Figure 11: Generated images and their nearest neighbors form the train set.

