# OpenReview forum: "In Defence Of Wasserstein"
_ICLR.cc/2024/Conference — Submitted to ICLR 2024_

### Official Review · Reviewer_hGFX · 2023-10-18

**Soundness:** 2 fair
**Presentation:** 2 fair
**Contribution:** 2 fair
**Rating:** 3
**Confidence:** 4

**Summary:**

This papers offers a new look into the "Do Wasserstein GANs approximate the Wasserstein distance" debate. The authors argue, despite previous litterature, that Wasserstein GANs actually __do__ minimize the Wasserstein distance. To justify this conceptual shift, they look at the form of the distance w.r.t. the choice of discriminator architecture. In this sense, the authors propose the idea of "patch Wasserstein distance", which is a Wasserstein distance between groups of patches. In their paper, the authors argue that, when the discriminator takes the form of a CNN, the WGAN minimizes this novel distance.

**Strengths:**

The authors tackle an important and relevant point in the intersection of Generative Modeling and Optimal Transport, that is, whether or not Wasserstein GANs actually minimize the Wasserstein distance. This problem is of key importance for the theoretical understanding of popular, optimal transport-based generative neural networks. The authors' writing is engaging, and they present their ideas and arguments in a fluid way.

**Weaknesses:**

__Major__

- __W1.__ The "Discrete Wasserstein GAN" corresponds to computing the (primal) Wasserstein distance between mini-batches sampled from the noise distribution, that is, $x\_{i} = f(z\_{i})$ and mini-batches sampled from the data set, that is, $y\_{i}$. However, such strategy was already described in two works: [Genevay, Peyré and Cuturi, 2018] and [Salimans et al., 2018]. Authors should include an appropriate citation to these works, and include a comparison with them in the discussion, highlighting, for instance, the effects of using exact OT (authors submission) in place of Entropic OT (previous work).

- __W2.__ The authors discussion is confusing w.r.t. the estimation of Wasserstein distances. For instance, Eqns. 1 and 2 refer to continuous OT, where $P$ and $Q$ are two distributions/measures. However, Eqn. 3 refers to a discrete OT, or, more formally, to an empirical estimate of the Wasserstein distance. __The authors could improve notation__. For instance, for (continuous) $P$ and $Q$, one has the true value of the Wasserstein distance $W_{p}(P, Q)$, which cannot be estimated directly since $P$ and $Q$ are unknown or too complicated. In contrast, when approximating $P$ and $Q$ empirically, one has indeed the discrete OT cost $W_{p}(\hat{P}, \hat{Q})$, based on sampled drawn from $P$ and $Q$. Note that this is an __empirical estimate__ (thus a random variable) and may be differ from the true $W_{p}(P, Q)$. With this remark in mind, authors should update their notation and discussion, making clear __what is minimized throughout GAN optimization__. Furthermore, the authors could consider the work of [Fatras et al., 2020], who explore in approximating OT through mini-batches. In my view, this play a major role in the learning of GANs (and any OT-base minibatch optimization algorithm for that matter).

- __W3.__ The theoretical results of the paper are a bit imprecise. On this point, I refer authors to my questions.

- __W4.__ Some notions introduced by the authors are never defined properly.

__Minor.__ Note, the minor points did not affect my overall score.

- Abstracts should be 1 paragraph long.

- __Notation.__ While using $W1$ for the Wasserstein distance is readable, authors should consider using subscripts (e.g., $W_{1}$). Also, for the sake of uniformity, authors should use $W_{1}$ along the paper (e.g., in Eqn. authors use $W(P, Q)$). Whenever authors discretize continuous OT, especially from samples, the authors should consider using "$\hat{\cdot}$", to highlight that the said quantity is an estimate rather than the true value.

__References__

[Genevay, Peyré and Cuturi, 2018] Genevay, A., Peyré, G., & Cuturi, M. (2018, March). Learning generative models with sinkhorn divergences. In International Conference on Artificial Intelligence and Statistics (pp. 1608-1617). PMLR.

[Salimans et al., 2018] Salimans, T., Zhang, H., Radford, A., & Metaxas, D. (2018, February). Improving GANs Using Optimal Transport. In International Conference on Learning Representations.

[Fatras et al., 2020] Fatras, K., Zine, Y., Flamary, R., Gribonval, R., & Courty, N. (2020, June). Learning with minibatch Wasserstein: asymptotic and gradient properties. In International Conference on Artificial Intelligence and Statistics (pp. 2131-2141). PMLR.

**Questions:**

- __Q1.__ The argmin in Eqn. 4 is taken w.r.t. which variable? Furthermore, I am curious if there is any relationship of this eqn. and the Barycentric Projection used in [Courty et al., 2017] (e.g., Eqn. 13 of the said reference)

__Related to Weakness 3.__

- __Q2.__ In __Theorem 2__ of the paper, the authors use "approximately minimizing" in their statement. On the paragraph following the proof the authors mention,

> The term ”approximately minimizing” in the theorem statement is the same as in the original proof of WGANs

Could you clarify to which proof the authors refer? Note that there are multiple theorems in the WGAN paper. Furthermore, I would like that the authors clarify what __approximately minimizing__ means. Does this comes from the fact of using mini-batches? Does this comes from the Lipschitzness constraint?

- __Q3.__ __Theorem 3__ is defined in loose terms, and is overall confusing. First, it is not clear how the distributions playing a role in this theorem are defined. Second, the authors seem to consider the "patch Wasserstein distance" between samples from underlying distributions. In this sense one would expect that the statement would hold w.r.t. a given probability according to the choice of samples (e.g., PAC theory), but the authors never offer any such analysis. An example on why this raises problems is with statements of the sort _"Since the number of patches is large in the M generated images and in the training set, and they come from the same distribution, then the Wasserstein distance over patches will be close to zero"_. Note that this statement is not informative since one does not know _how close to zero_ the said distance gets. A similar remark can be made about the second sentence, _"On the other hand, when we take the linear combination of $M / N$ different images we will almost always create additional patches that are not one of the $K$ possible patches in the distribution"_. Here, even if we agree that the linear combination operation will generate patches not in the possible set of patches (and that is not guaranteed, as it depends on the set of patches being combined!), the affirmation that this occurs _almost always_ is very loose. Again, it should be defined as a probability over the choice of samples.

---

> ### Author Response · Authors · 2023-11-15
>
> Thank you for your very detailed review of our work. We agree with some of the concerns that you raised but reading your first two weakness points lead us to believe you may have misunderstood the discrete generative settings we investigate throughout the paper. We would sincerely appreciate it if you read our detailed response and tell us if this somehow sheds another light on our work.
>
> ## Additional comments:
> - __Weakness 1:__ The papers you referred to solve the OT problem between sampled mini batches as an approximation of the OT between the entire data and generated distributions. As these papers show, approximating W1 over distributions by W1 over samples is dangerous and can lead to wrong conclusions. It is precisely for this reason that we analyze a scenario where both generated and data distribution are discrete and therefore we can solve the OT between the entire sets without resorting to samples. Specifically the OTmeans algorithm solves the OT problem between the M generated images and the entire dataset in each iteration.
>
> - __Weakness 2:__ As in our last answer, we suspect there is a misunderstanding of what DiscreteWGAN means. We never use samples to approximate the distributions empirically as the distributions are already discrete ones. This allows us in figure 2 to plot exact W1 and not approximations to W1 that are based on samples as was done in previous works.
>
> - __Weakness 3-4:__ We agree that Theorem 3 and its proof are written too informally and we will rewrite them in our final revision. See answer to questions below.
> - __Question 1:__ The argmin should be w.r.t $x_j$ we will rewrite this in our final revision with a slack variable ‘z’. We did not mean to specifically address Barycentric Projections .We would appreciate it if you direct us to the full reference you mentioned as we did not find it.
>
> - __Question 2:__ As you know, the WGAN iterative training procedure is claimed to minimize  the data/generated W1 distance only if
>     1. The discriminator family of functions defined by the dicsriminator’s architecture is big “enough”.
>     2. The discriminator is trained to optimality before each generator step.
>     3. The approximation of the expectation of Kantorovich potentials in the dual form of W1 with minibatches is accurate.
>     4. The Lipshitz constraint on the discriminator is held.
>
>     Even then the SGD optimization may end in a local minima.
>     All these conditions should be held whether we are dealing with W1 between images or patches. Conditioned 1-2 are detailed in proposition-1 in the original GAN paper [1] and conditions 2-3 are detailed throughout the WGAN paper [2]. Of course we will find a way to make this claim clearer in our final revision.
>
> - __Question 3:__ This seems again to be based on the same misunderstanding above. We never use samples to approximate W1 but rather can compute it exactly since it is discrete. We need M,N to be large to avoid quantization errors, not for statistical efficiency. Consider the following simple example. Suppose we have (N=100) binary variables of which 60 are 1 and 40 are 0. If we have (M=10) binary variables of which 6 are 1 and 4 are 0, we can get 0 Wasserstein distance. But if we have (M=11) binary variables, then we cannot get 0 Wasserstein distance due to quantization effects. When both M and N are large, these quantization effects can be neglected.
>
> ## Citations:
> [1] Goodfellow, Ian, et al. "Generative adversarial networks." Communications of the ACM 63.11 (2020): 139-144.
>
> [2] Arjovsky, Martin, Soumith Chintala, and Léon Bottou. "Wasserstein generative adversarial networks." International conference on machine learning. PMLR, 2017

---

> > ### Comment · Reviewer_hGFX · 2023-11-17
> > **Response to Authors**
> >
> > Dear authors,
> >
> > First I would like to acknowledge and apologize for my error. Indeed, I understood the principle behind DiscreteGAN wrongly.
> >
> > Second, I indeed forgot the full reference of [Courty et al., 2017]. Here it is,
> >
> > > N. Courty, R. Flamary, D. Tuia and A. Rakotomamonjy, "Optimal Transport for Domain Adaptation," in IEEE Transactions on Pattern Analysis and Machine Intelligence, vol. 39, no. 9, pp. 1853-1865, 1 Sept. 2017, doi: 10.1109/TPAMI.2016.2615921.
> >
> > The equation I mentioned in my initial commentary is eq. 13.
> >
> > Furthermore, I am still not sure on how you measure the Wasserstein distance between $P$ and $Q$ exactly. For instance, in the generative modeling literature, one usually makes the hypothesis that the training set $\set{y\_{i}}\_{i=1}^{N}$ comes from an underlying continuous distribution $P$ from which $y_{i}$ is sampled. In this sense, the generated distribution $Q_{\theta}$ is the parametric distribution (parametrized by $\theta$) that better approximates the data/underlying distribution $P$.
> >
> > Given this motivation, in my view while one may fix the samples from the generated distribution $Q_{\theta}$ and optimize w.r.t. $\theta$, one still has a sampling effect on $P$. At worst case, considering fixed $N$ points from $P$ is still a (rather big) sample from the underlying distribution.
> >
> > Can the authors provide further discussion and motivation over these two points?
> >
> > __Note.__ I just found an inconsistency in page 3. Before presenting the OTMeans algorithm, the authors define $P$ as the generated distribution, whereas in theorem 1 $P$ is the training distribution.

---

> ### Author Response · Authors · 2023-11-19
>
> Thank you for the reference and additional comments
> - Indeed equation 4 in our paper is equivalent to the barycentric mapping of the OT solution found on each teration of the OTMeans algorithm. Note that we did not use any general property of barycentric mappings and simply used the fact that for L2 norm the solution is a weighted average of “target” points. As noted in the text that follows eq. 13 in the paper you referenced, when the cost function is the squared l2 metric the barycentric mapping is the weighted average of all targets. We used the (non-squared) L2 loss for which the barycentric mapping is still a weighted average of the targets but the finding the weights requires using iterative reweighed least squares as mentioned in our paper.
> - When fitting parametric statistical models to data, it is common to consider three distributions. $P_{\theta}$ is the parameterized distribution that we are fitting, $Q_{data}$ is the empirical distribution, and $Q$ is the distribution from which the data was sampled. You are certainly correct that ideally we want $P_{\theta}$ to be as close as possible to $Q$ but of course we do not have access to it during training and so we resort to making $P_{\theta}$ as close as possible to $Q_{data}$. See for example page 109 of the textbook “Probabilistic Machine Learning “ by Kevin Murphy (draft available at: https://github.com/probml/pml-book/releases/latest/download/book1.pdf) which shows that Maximum Likelihood is equivalent to minimizing the $KL$ distribution between $P_{\theta}$ and $Q_{data}$. Just as a method for maximizing the likelihood is evaluated by how well it minimizes  $KL(P_{\theta}, Q_{data})$, a method for minimizing the Wasserstein distance should be evaluated by how well it minimizes $W1(P_{\theta}, Q_{data})$. In our setting, both $P_{\theta}$ and $Q_{data}$ are discrete so that we can evaluate $W1(P_{\theta}, Q_{data})$ exactly. We will fix all $P/Q$ data/generated notation inconsistencies in our final revision of the paper.

---

> > ### Comment · Reviewer_hGFX · 2023-11-20
> > **Response to Authors**
> >
> > Dear authors,
> >
> > Thanks again for the extended discussion.
> >
> > Even though at first I misunderstood the principle of your contribution, I had other concerns about the precision of your theoretical results. Based on my remarks, and those of other reviewers (e.g., Reviewer L98k), these concerns were not fully addressed during this phase of discussion. As a result, I keep my initial score.
> >
> > Overall, I think the authors consider an important and interesting problem, but the theoretical results need to be defined in more precise terms. On top of that, I think further theoretical discussion on considering discrete distributions should be added to the main paper.

---

> > > ### Author Response · Authors · 2023-11-22
> > >
> > > Thank you again for your time reviewing our work. We would appreciate it if you clarified what you mean by " the theoretical results need to be defined in more precise terms."  so that we could improve the writing in the next version.

---

### Official Review · Reviewer_L98k · 2023-10-23

**Soundness:** 2 fair
**Presentation:** 3 good
**Contribution:** 2 fair
**Rating:** 3
**Confidence:** 3

**Summary:**

This paper proposes to analyze an intriguing property of Wasserstein GANs, which suggests that their success may be attributed to a partial failure in minimizing the Wasserstein distance. The authors demonstrate that, when applied to image data, the characteristics of the generated images are, in fact, influenced by the architecture of the discriminator. They use a discrete setting, where the generator takes as input a vector from a set of M vectors. Their study reveals the following findings:
1. WGANs with fully connected discriminators do effectively minimize the Wasserstein distance.
2. WGANs with CNN+GlobalAvgPool discriminators minimize a Patch-based Wasserstein distance.
3. WGANs with CNN discriminators minimize location-specific patch Wasserstein distance.

**Strengths:**

* The authors analyze and showcase the impact of the discriminator's architecture and inductive bias on the images generated by a WGAN. Doing so, they partly answer the question of what do WGANs actually minimize, since there are suspicions that they do not minimize the Wasserstein distance.

* The paper is well written and easy to read.

**Weaknesses:**

* Missing an important reference [1] which tackles and solves the same question. They prove theoretically that WGANs-GP actually correspond to the minimization of a "congested transport distance" rather than the standard Wasserstein distance. The authors claim that it explains the diversity observed in generated distributions, since this distance penalizes "congestion" in transport plans. What can you comment on this?
[1] Tristan Milne and Adrian I. Nachman. "Wasserstein gans with gradient penalty compute congested transport." Conference on Learning Theory. PMLR, 2022.

* What about WGANs with transformer architectures, which also generate sharp images although they have no inductive bias for patch learning? Or other architectures like MLP-Mixer? Notably, TransGAN [2] uses a WGAN-GP loss and still generates high-quality images.

[2] Jiang, Y., Chang, S., & Wang, Z.. Transgan: Two pure transformers can make one strong gan, and that can scale up. NeurIPS 2021.

* About the interpretation of CNN-GAP and Theorem 2: Depending on the depth of the neural net and kernel size of convolutional filters, the receptive field of each patch might actually be the whole image. Then, why would it lead to different results than the fully connected discriminator? Wouldn't it be another reason, like optimizability of the neural net, explaining the fact that we have different results?  To verify this, it could be interesting to progressively increase the size of receptive field, and observe the changes in the generated images.

* About the setting $M<<N$ : is it really realistic when GANs actually try to approximate a discrete distribution (empirical distribution with N training points) with a continuous one (generator with continuous latent space)?  How to connect the results with $M<<N$ to the realistic setting? What does it say about the realistic setting? This is not clear from the paper.

* "As is common in WGAN training we do not train the discriminator until convergence but rather do a single update after each mini-batch and use the gradient penalty to regularize the generator" Actually, in WGANs, discriminators are generally either updated for several steps per generator update, either have a larger learning rate than generators (TTUR). This might lead to a problem in your setting (2.1). If the discriminators was trained optimally, we could expect the generator to overfit and to reproduce some images of the training set. How come does it not happen? Is it because of your setting M<<N, or could it happen because of suboptimal optimization?

* The multi-scale minimization of SWD seems interesting but should be studied more in depth. Moreover, it should be explained more carefully, at least with some equations or with an algorithm.

* Minor remarks on writing/formatting problems: title, some citations are not well included in the text (page 3: "the POT package Flamary et al. (2021)" or "reweighted least squares. Weiszfeld (1937)"), no end to the proof of Theorem 2, 'locaton' -> 'location' page 8, "in defence of wasserstein" in the Openreview title.

**Questions:**

See weaknesses.

---

> ### Author Response · Authors · 2023-11-15
>
> Thank you for your very detailed review of our work. Below are our detailed responses to the weaknesses and questions you raised. We agree with many of your concerns but we believe that they do not detract from our main result. To the best of our knowledge, we are the first to show an analytical connection between the loss optimized by GANs and the discriminator architecture and to show that minimizing patch Wasserstein gives sharp images while minimizing image Wasserstein gives blurry images. We encourage you to read our answers and let us know if they are satisfactory and if so, to consider revising your rating.
>
> ## Additional comments:
> - __Weakness 1:__ We agree that the work of [1] is very relevant to our work as it too, tries to answer the question raised in the same papers we reference. However it’s worth noting that the gradient penalty is not essential to the success of WGANs nor to  any of our theoretical results. WGANs also generate sharp images with other regularizers such as weight clipping or spectral normalization so that [1] cannot explain their success. More importantly [1] does not address the influence of the architecture on the minimized distance which we have shown is crucial to understanding the success of WGANs.
> - __Weakness 2:__ Transformer architectures are indeed interesting but we disagree with the statement that “they have no inductive bias for patch learning”. In fact, the key to the success of visual transformers is that they use image patches as tokens and many visual transformers used as classifiers give the same classification to an original image and an image in which the patches have been randomly permuted (see [3]). Thus our theorem 2 can be extended to prove that WGANs also minimize patch Wasserstein distance when using such visual transformers.
> - __Weakness 3:__ The size of the receptive field  is an interesting point we referred to shortly at the end of section 6 of the paper. We have actually conducted the experiment that you suggest with different receptive field sizes in the discriminator and found that the generated images change as expected by the theory. We will include these in the final version. Indeed many actual CNN architectures have a very big potential receptive field but  please see [2] where it is shown that the “effective” receptive field of commonly used CNNs is much smaller than its potential size.
> - __Weakness 4:__ Regarding the continuous generator, please see the second paragraph of the “limitations and extensions” section and our comment to all reviewers.
> - __Weakness 5:__ Training a GAN requires a delicate equilibrium between the two trained players, there is no secret ingredient to finding it and while usually TTUR or more discriminator updates are required this may change according to the used architecture. As you can see in our supplementary code we did use TTUR for most of our experiments. As for the second part of your comment: We did not train the discriminator to optimality. Hypothetically one should expect some optimization difficulties and even if the discriminator is overfit, it is always overfit with respect to the current generator which can easily deceive the discriminator in many non-natural ways on the next step. So we should not necessarily expect to see it generating copies of some of the dataset.
> - __Weakness 6:__ Our multiscale SWD optimization is very straightforward and only requires changing the patch size we use throughout training. We will add more details in our final revision.
> - __Weakness 7:__ Thank you for the formatting suggestions that we will correct in our final revision
>
> ## Citations:
> [1] Tristan Milne and Adrian I. Nachman. "Wasserstein gans with gradient penalty compute congested transport." Conference on Learning Theory. PMLR, 2022.
>
> [2] Brendel, Wieland, and Matthias Bethge. "Approximating cnns with bag-of-local-features models works surprisingly well on imagenet." arXiv preprint arXiv:1904.00760 (2019).
>
> [3] Qin, Yao, et al. "Understanding and improving robustness of vision transformers through patch-based negative augmentation." Advances in Neural Information Processing Systems 35 (2022): 16276-16289.

---

> > ### Comment · Reviewer_L98k · 2023-11-20
> >
> > I acknowledge your answers. Especially, I appreciate your answers about my concerns on 1) **the architecture**: your point is valid that there is still a bias for patch learning in transformers  ; 2) **the receptive field**: even though the receptive field is large, the effective receptive field might be according to Wieland and Bethge, 2017.
> >
> > However, I am still concerned that some experiments are lacking a rigorous demonstration of the proposed assertions.  1) About the **receptive field**: while the depth might not be useful to increase the receptive field, what about the kernel size of the convolutional filters? modern CNNs have shown increased capacity while using very large kernel sizes. 2) About the setting **M << N**: despite acknowledging the new figure presented by the authors, I would expect a more careful study of this issue.
> >
> > Therefore, at present, I do not consider increasing my rating.

---

> > > ### Author Response · Authors · 2023-11-22
> > >
> > > Thank you for your comment.
> > >
> > > Can you please clarify what you mean by "experiments are lacking a rigorous demonstration of the proposed assertions." ?
> > >
> > > Our assertions about what WGANs are minimizing are primarily based on  theorems, not on experiments. For example, the assertion that WGANs with CNN-GAP discriminator minimize the patch Wasserstein distance is based on Theorem 2, the assertion that WGANS with CNN-FC discriminator minimize a location-specific patch Wasserstein distance is based on Theorem 4 and so on. We provide the experiments to make it easier to see what the theorems assert, but we don't think the experiments by themselves can be a "rigorous demonstration" of the assertions.

---

### Official Review · Reviewer_ZrfD · 2023-10-27

**Soundness:** 3 good
**Presentation:** 4 excellent
**Contribution:** 3 good
**Rating:** 8
**Confidence:** 3

**Summary:**

The paper contributes to a debate "do Wasserstein GANs actually optimize Wasserstein distance and is that why they work well?"

It's previously been noted that a batch of blurry Kmeans images achieve better Wasserstein distance than a batch of real images. It brought up a question if W1 is a sensible optimization metric for generative image modeling. This paper analyzes the influence of discriminator architecture on the exact mathematical objective. The authors claim that a fully connected discriminator indeed optimizes image Wasserstein distances and yields blurry images in agreement with the prior work. However, convolutional architectures optimize Wasserstein distance on the distribution of image patches which is different and doesn't lead to blurry images. Moreover, the authors compare CNN discriminator with Global Average Pooling layer vs FC layer and notice that it determines whether the GAN captures location information (ex: where eyes should be on human faces).

It is worth noting that the authors use a discrete version of WGAN in order to faciliate the exact computation of Wasserstein distance: they fix a certain number of latent vectors.

The authors also suggest a simple way to directly optimize this simplified version of Wasserstein distance. It supports their claims of the influence of discriminator objective on the generated image distribution.

**Strengths:**

Related work is great, it explains well the prior conversation about WGANs and why there are doubts whether it is a good thing to optimize the Wasserstein distance. The experiments are quite sound and easy to follow. The paper is well-written in general. I also liked that the authors support the WGAN experiments with alternative ways to directly optimize Wasserstein distance. I think the idea is also original enough to warrant a publication and will be useful for future research even if it doesn't have immediate practical implications like pushing SOTA.

**Weaknesses:**

I suppose the main weakness is that the architectures analyzed in the paper are quite dated and simple compared to what people use nowadays. The datasets are pretty basic and small as well. The field moved a lot since the WGAN paper hence most likely their findings can't be applied to the state-of-the-art GANs. Nevertheless, I find this analysis interesting and insightful, it highlights that the architecture influences the exact mathematical function we optimize during training and affects the outcome in a more dramatic way than just "this network gets better accuracy on ImageNet-like problems".

**Questions:**

I suggest to add network architecture diagrams at least in supplementary material. Focusing on subtle differences in architectures without clearly explaining them is odd.

---

> ### Author Response · Authors · 2023-11-15
>
> Thank you for your very detailed review of our work. We really appreciate your attention to the importance of our research subject and its significance to future research. We believe that GANs are still widely used and their underlying mechanism is massively misunderstood. Understanding and simplifying it is a very important task in our view. We will be happy to answer any further questions.

---

### Official Review · Reviewer_scef · 2023-11-01

**Soundness:** 2 fair
**Presentation:** 2 fair
**Contribution:** 2 fair
**Rating:** 3
**Confidence:** 4

**Summary:**

This paper claims that the WGANs optimizes the Wasserstein distance between the real data and generated data distributions. The authors used a so-called discrete generator which generates data from a finite number of fixed random noises. They claimed that when the discriminator architecture is a fully connected network, then WGANs minimize the Wasserstein distance between real data and generated data distributions. If the discriminator architecture is a Convolutional Neural Network (CNN), then the WGANs minimize the Wasserstein distance between the patches in the real images and generated images.

**Strengths:**

The authors showed in Theorem 3 in this paper that when a discriminator is a CNN with global average pooling in the second last layer, the WGAN approximately optimizes the Wasserstein distance between patches of the real images and the generated images.

**Weaknesses:**

Weakness:

1. The theoretical result in Theorem 1 does not support the claim that when the discriminator architecture is a fully connected network, WGANs minimize the Wasserstein distance between real data and generated data distributions.
    - My major concern is the discrete generator, that generates only $M$ samples. In fact, a generator could generate infinity number of samples. So, "when each generated sample is a linear combination of at least $N/M$ training samples" in Theorem 1 is not correct. Therefore, why previous works generates blurry images cannot be explained by the linear combinations of training samples introduced in Theorem 1.
    - The authors obtained Theorem 1 from Eq. 4, which is a primal form of Optimal Transport (OT), but they claimed that WGANs produce the results in Theorem 1. In fact, a WGAN approximates the OT in the dual form, and the generator and discriminator in a WGAN perform updates alternatively. The finally generated data may be different from the generated data $x_j$ shown in Eq. 4. Therefore, Theorem 1 does not support the claim that when the discriminator architecture is a fully connected network, WGANs minimize the Wasserstein distance between real data and generated data distributions.

2. The premises in Theorem 3 does not hold.
    - The authors assume $M < N$ in Theorem 3, but as I mentioned above, $M$ should be infinity or should be at least much greater than $N$. Also, the authors should assume in Theorem 3 that the patch size of the generated images and the real images should be the same.

3. WGANs have been proposed for many years. I cannot imagine that a WGAN generates poor images like those shown in Figs. 3 and 5. The authors should choose CNNs for the discriminator and generator.

4. The authors uses the Sliced Wasserstein Distance (SWD) to approximate the Wasserstein distance between all the 16x16 patches. How close is a SWD to the real Wasserstein distance? Are there any evidences of literatures that indicating SWD and Wasserstein distance are consistent?

5. Writing in several parts is not clear.
    - The proof of Theorem 4, when $w_{jc} = 0$ for all $j \ne i$. Here $i$ and $j$ are just two random indices. I don't understand when $j \ne i$.
    - $x_j = y_i$ in theorem 1 is not correct.

**Questions:**

In the end of the first paragraph of Sec. 2.1, the authors use "gradient penalty to regularize the generator." Is this a typo? Because, in general, we use the gradient penalty to regularize the discriminator.


Do the patches have overlap in Theorems 2 and 4?


Typos:

"in the layer last linear layer " in Theorem 4.

---

> ### Author Response · Authors · 2023-11-15
>
> Thank you for your very detailed review of our work. We are concerned with some parts of your review which suggest some misunderstanding of the paper’s claims in theorem 1 and the experiments following it. We would sincerely appreciate it if you read our detailed response and reread the paper and let us know if we can clear up further misunderstandings.
>
> Nowhere in the paper do we claim that the theoretical results in theorem 1 “support the  claim that when the discriminator architecture is a fully connected network, WGANs minimize the Wasserstein distance between real data and generated data distributions.”. Theorem 1 simply characterizes the optimal solution to the Wasserstein distance minimization problem when the generator is discrete. It says nothing about WGANs.
>
> Because we are using a discrete generator, we can exactly compute the Wasserstein distance between any M generated images and the N training images. In figure 2, we plot this exact Wasserstein distance and show that for all the datasets, the solution found by WGANs with a FC discriminator (blue curve) is close to the optimal solution that is found using OT-means (green curve) . Furthermore, the images that are found by WGANs have the same characteristics that the optimal solution has (they are blurred, rather than sharp as predicted by theorem 1). We believe that this is strong evidence that the WGANs with an FC discriminator indeed minimize the Wasserstein distance.
>
> We agree that our analysis is for the case of a discrete generator and this is a simplification that we explicitly introduced in page 3 of the paper in order to allow us to compute exact Wasserstein distances. We think it is unfair to say in your review  that “our premises do not hold”  or that “M should be infinity”. A WGAN with a discrete generator is still a WGAN and please note that ALL of the experiments we show in the paper are with a discrete WGAN generator. Yet the same discrete generator can generate blurry images (when the discriminator is FC) or sharp images (when the discriminator is convolutional). That is what we are trying to explain and we believe that our analysis is the first to provide an explanation.
>
> We explicitly discuss how the results of our paper relate to WGANs with a continuous generator in the second paragraph of the “limitations and extensions” section and in the comment to all reviewers.
>
> ## Additional comments in response to your questions:
>
> - __Weakness 3:__ SOTA WGANs are indeed capable of generating much prettier images and they mostly use CNN generators. However, our goal here was to understand how GANs work and as with many theoretical analysis papers we use simpler architectures in order to get to the crux of the GAN mechanism. Moreover, there are no theoretical requirements that GAN should only work with a specific architecture and in fact, the original GAN paper [1] used the same FC architecture and the original WGAN paper [2] uses both FC and CNNs.
>
> - __Weakness 4:__ As far as we know, there is no direct connection between SWD and W1 distance apart from the definition of SWD as W1 over 1-dimensional projections of the data. They are both metrics and achieve zero if and only if the distributions are equal. We agree that this fact should be noted if we describe SWD as a “proxy” to the W1 distance.
>
> - __Weakness 5:__
>     - We agree that both notations are quite confusing and we will fix them in our final revision
> We abused the index $i$ here: When trying to define the function class $CNN_k$ for some index $k$ we define it using eq. 8 where $W_{jc}=0$ $\forall j \neq k$.
>     - The term $x_i=y_j$ will indeed be replaced by $\forall i \in [N] x_i=y_i$
>
> - __Question 1:__
>     - Of course, we meant that the gradient penalty regularizes the discriminator and not the generator.
>     - Overlap between patches should not affect the theoretical results as it will only change the distribution we are dealing with. In practice the CNN architecture we used, like most CNNs, uses convolutions with stride=2 thus the stride of the receptive field is $2^n$ where $n$ is the number of convolutions. For example in the CNN+GAP architecture we used, there are 3 convolutions with stride 2 and hence the stride of the receptive field is 8.
>
> ## Citations:
> [1] Goodfellow, Ian, et al. "Generative adversarial networks." Communications of the ACM 63.11 (2020): 139-144.
>
> [2] Arjovsky, Martin, Soumith Chintala, and Léon Bottou. "Wasserstein generative adversarial networks." International conference on machine learning. PMLR, 2017.

---

> > ### Comment · Reviewer_scef · 2023-12-03
> > **Response to authors**
> >
> > Thanks the authors for the response!
> >
> > I think the writing of the text below Eq. (2) in the bottom of page 2 is misleading. It confuses me that the $W(P, Q)$ in Eq. 2 is the WGAN objective. Also, Theorem 1 mentioned that $W(P, Q)$ is obtained when each generated sample is a linear combination of at least $N/M$ training samples.
> >
> > In addition, the authors mentioned in the abstract that "when the discriminator is fully connected, standard WGANs indeed minimize the Wasserstein distance between the generated images and the training images". This is a strong claim that is not supported in any theorems in this paper, including Theorem 1. The authors just showed some experiments in Fig. 2, which I do not think they are sufficient to support this strong claim.
> >
> > Still, from the response, I am not convinced by using the discrete generator for the analysis in this paper, as most of my critical concerns are related to this discrete generator.

---

### Author Response · Authors · 2023-11-15

Some of the reviewers raised concerns about the relevance of the discrete WGAN to regular GANs with continuous priors. As we wrote in the text, we explicitly use a discrete WGAN in order to enable exact computation of the Wasserstein distance and to allow us to prove results on the properties of the minimizers. Nevertheless, a discrete WGAN is still a WGAN and our results clearly show that discrete WGANs will approximate a different Wasserstein distance depending on the classifier architecture.

In order to show that the same behavior holds for continuous WGANs we uploaded a figure (link below) that compares the discrete WGAN outputs from figures 2 and 3 of our paper to the results of the same WGAN when the prior is continuous normal distribution (M=$\infty$). As you can see the outputs look similar even with the normal prior. Specifically, when the discriminator is FC the outputs still look like minimizing the Image Wasserstein distance and when the discriminator is CNN+GAP the outputs looks like minimizing the patch Wasserstein distance.

The reason that the continuous WGAN behaves like this is briefly referred to in the second paragraph of the “limitations and extensions” section of our paper: Even when using a continuous prior the generator has a finite capacity and is able to generate only so many different outputs. This capacity is determined by its architectures and number of parameters. We preferred to study the explicit capacity constraint of the discrete WGAN but as our results show, the behavior of the continuous WGAN is similar.

Link to figure: https://ibb.co/Th0bNy8

---

### Author Response · Authors · 2023-11-23

In reading the discussion, it seems that several reviewers are under the impression that our analysis only holds for discrete WGANs.

In fact, our main theoretical result (that WGANs with a convolutional discriminator minimize a patch Wasserstein distance) holds for any generator. This can be seen by the statement and the proofs of Theorems 2 and 4. It is only theorems 1 and 3, which characterize the optimal solutions of the Wasserstein minimization problem that require discreteness of the generator.

---

### Meta-Review · Area_Chair_uav4 · 2023-12-06

**Metareview:**

The paper has been acknowledged for its interesting exploration of the properties of Wasserstein Generative Adversarial Networks. The authors' rebuttal was found to be strong, particularly in clarifying the role of the architectures and their relationship to relevant prior works. However, most reviewers continue to express concerns regarding the relevance of employing a discrete generator in this context. While the addition of a new figure in the rebuttal was well-received, the consensus among reviewers is that this alone does not sufficiently address their reservations. Furthermore, the theoretical claims presented in the paper require greater clarity. This aspect, in particular, necessitates substantial rewriting and revision to ensure that the theoretical underpinnings are conveyed more effectively.

**Justification For Why Not Higher Score:**

I think the contribution are not strong enough, and the studied setting (discrete setup) not very relevant.

**Justification For Why Not Lower Score:**

N/A

---

### Decision · Program_Chairs · 2024-01-16

Reject